# HIV Knowledge and Stigmatizing Attitude towards People Living with HIV/AIDS among Medical Students in Jordan

**DOI:** 10.3390/ijerph19020745

**Published:** 2022-01-10

**Authors:** Malik Sallam, Ali M. Alabbadi, Sarah Abdel-Razeq, Kareem Battah, Leen Malkawi, Mousa A. Al-Abbadi, Azmi Mahafzah

**Affiliations:** 1Department of Pathology, Microbiology and Forensic Medicine, School of Medicine, The University of Jordan, Amman 11942, Jordan; ma.alabbadi@ju.edu.jo (M.A.A.-A.); mahafzaa@ju.edu.jo (A.M.); 2Department of Clinical Laboratories and Forensic Medicine, Jordan University Hospital, Amman 11942, Jordan; 3Department of Translational Medicine, Faculty of Medicine, Lund University, 22184 Malmö, Sweden; 4School of Medicine, The University of Jordan, Amman 11942, Jordan; aly0174238@ju.edu.jo (A.M.A.); sar0191156@ju.edu.jo (S.A.-R.); kry0184991@ju.edu.jo (K.B.); Lyn0194163@ju.edu.jo (L.M.)

**Keywords:** discrimination, HIV-related stigma, HIV/AIDS, college students, HIV knowledge, education, Middle East, MENA, prejudice, medical education

## Abstract

The stigmatizing attitude towards people living with HIV/AIDS (PLWHA) can be a major barrier to effective patient care. As future physicians, medical students represent a core group that should be targeted with focused knowledge and adequate training to provide patient care without prejudice. The aim of the current study was to examine HIV/AIDS knowledge, and the stigmatizing attitude towards PLWHA, among medical students in Jordan. The current study was based on a self-administered online questionnaire, which was distributed during March–May 2021, involving students at the six medical schools in Jordan, with items assessing demographics, HIV/AIDS knowledge, and HIV/AIDS stigmatizing attitude, which was evaluated using the validated HIV-stigma scale. The total number of respondents was 1362, with predominance of females (*n* = 780, 57.3%). Lack of HIV/AIDS knowledge among the study participants was notable for the following items: HIV transmission through breastfeeding (40.8% correct responses), HIV is not transmitted through saliva (42.6% correct responses), and vertical transmission of HIV can be prevented (48.8% correct responses). Approximately two-thirds of the respondents displayed a positive attitude towards PLWHA. For six out of the 14 HIV/AIDS knowledge items, lack of knowledge was significantly correlated with a more negative attitude towards PLWHA. Multinomial regression analysis showed that a significantly more negative attitude towards PLWHA was found among the pre-clinical students compared to the clinical students (odds ratio (OR): 0.65, 95% confidence interval (CI): 0.43–0.97, *p* = 0.036); and that affiliation to medical schools that were founded before 2000 was associated with a more positive attitude towards PLWHA compared to affiliation to recently founded medical schools in the country (OR: 1.85, 95% CI: 1.42–2.42, *p* < 0.001). About one-third of medical students who participated in the study displayed a negative attitude towards PLWHA. Defects in HIV/AIDS knowledge were detected for aspects involving HIV transmission and prevention, and such defects were correlated with a more negative attitude towards PLWHA. It is recommended to revise the current medical training curricula, and to tailor improvements in the overall HIV/AIDS knowledge, which can be reflected in a more positive attitude towards PLWHA, particularly for the recently established medical schools in the country.

## 1. Introduction

More than forty years have passed since the first reports on acquired immunodeficiency syndrome (AIDS); nevertheless, human immunodeficiency virus (HIV) infection continues to pose a major global health issue [1]. Besides the health aspects that face people living with HIV/AIDS (PLWHA), social and psychological problems are considered important challenges as well [2]. In particular, stigmatization and discrimination represent major barriers leading to poor access to healthcare services and testing, along with a reduced adherence to treatment among PLWHA [3]. Thus, studying the scope of stigma and discrimination against PLWHA is considered among the top priorities for HIV/AIDS research [4,5].

The concept of HIV stigmatization can be defined as the negative attitudes and beliefs directed towards PLWHA [6,7]. Such negative attitudes entail prejudice and discriminatory behavior towards PLWHA, resulting in enormous forms of damage among them [2]. The aforementioned definition of HIV stigmatization can appear simple considering the complex nature and high diversity of this phenomenon, which varies in different cultural settings [8,9].

The negative impact of the stigmatizing attitude towards PLWHA extends beyond issues related to treatment and testing to involve other aspects, such as reduced linkage to healthcare services [10]. Subsequently, HIV stigma can undermine the efforts needed in the management and prevention of the disease [11,12]. Additionally, potential consequences of HIV stigma can involve less willingness to disclose HIV status, negative impact on health seeking behavior, and negative impact on the needed social support [13,14].

The study of stigmatizing behavior directed towards PLWHA among medical students can be valuable to reveal the defects and gaps in medical education and clinical training courses [15,16,17]. This can be related to the previous evidence showing that lack of adequate skills and improper training might be associated with fear and avoidance, and subsequent negative attitude [18,19]. Consequently, this can end up as a negative impact on patient care [20]. Specific HIV knowledge gaps that can result in a stigmatizing behavior include the inaccurate notions on how the virus is transmitted [13,21,22]. The misconceptions such as the belief that HIV can be transmitted by handshaking, hugging, or sharing utensils, can lead to fear of virus acquisition, and subsequent stigmatizing behavior [23,24].

The study of different aspects in relation to HIV/AIDS can have important implications in the Middle East region [25]. Contrary to the general perception that HIV/AIDS does not represent a public health issue in the region, a previous study indicated that HIV infection spreads locally in the Middle East and North Africa (MENA) region at a high rate [26]. In addition, the MENA region witnessed an increase in HIV infections since 2010 by 21%, in spite of the global reduction in HIV cases [27]. Moreover, the sexual nature of HIV transmission can be viewed as a cultural taboo topic in a majority of Middle East countries including Jordan [28,29]. Thus, such culturally sensitive topics can result in less orientation and discussion—even within the medical field—with subsequent lack of knowledge about sexually transmitted infections (STIs) and stigmatization towards the affected groups [30,31,32]. This lack of knowledge has been demonstrated in recent studies involving HIV and the human papillomavirus (HPV), which is another causative agent of STIs [33,34,35,36]. The concentration of HIV infection among marginalized, and often neglected populations (e.g., female sex workers (FSWs), men who have sex with men (MSM), injection drug users (IDUs), and prisoners) further augments the problem of stigmatization and discrimination [37,38,39].

In the current study, the main objectives were to assess the overall HIV/AIDS knowledge (particularly in aspects involving transmission and management) among medical students in Jordan, and to evaluate the overall stigmatizing attitude of medical students towards PLWHA.

## 2. Materials and Methods

### 2.1. Study Design

This cross-sectional study was based on a self-administered online questionnaire that was created using Google Forms. The questionnaire was designed to assess HIV/AIDS knowledge and the stigmatizing attitude among medical students in Jordan. The survey was distributed between 25 March 2021 and 5 May 2021 using a convenient sampling approach starting with the contacts of the authors (four of whom were medical students at the University of Jordan), who distributed the survey on Facebook and WhatsApp through posting the survey link on the major pages and groups targeting medical students in Jordan. The online questionnaire targeted students who were affiliated to the six medical schools currently present in Jordan. Participation was voluntary without any forms of incentives to those who responded.

In Jordan, the latest reports on the number of medical students who were enrolled in the six medical schools at the following universities: the University of Jordan (UJ), Jordan University of Science and Technology (JUST), Mutah University (MU), the Hashemite University (HU), Yarmouk University (YU), and Al-Balqa Applied University (BAU), was about 10,000 as of the end of curriculum year 2020/2021 (the Ministry of Higher Education, Jordan, personal communication). Since the official language of studying at the medical schools in Jordan is English, it was used in survey construction and distribution. Two medical schools in Jordan were established in the last century (1971 for UJ and 1986 for JUST), while the remaining four schools were founded in the current century (2001 for MU, 2006 for HU, 2013 for YU, and 2015 for BAU). Studying medicine in the six medical schools in Jordan involves six academic years; the first three years involve pre-clinical education, while the last three years involve clinical education and training. The minimum required sample size was estimated at 965 respondents considering a total number of 10,000 medical students in Jordan, and a margin of error equaling 3.0% (with 95% confidence interval) [40].

### 2.2. Ethical Permission

The study was approved by the Department of Pathology, Microbiology, and Forensic Medicine and by the Scientific Research Committee at the University of Jordan, School of Medicine (reference number: 1310/2021/67; March 2021). An informed consent was ensured by the presence of an introductory section of the survey used in this study, with a mandatory question asking for agreement from the respondent to participate in this survey study. All collected data were treated with confidentiality according to the guidelines of Helsinki declaration.

### 2.3. HIV Knowledge Survey Items

The response to all survey items was mandatory for successful submission and participation in the study. Assessment of the general characteristics of the study respondents was conducted using six items (age, sex, nationality (Jordanian vs. non-Jordanian), medical school (older: UJ or JUST vs. more recent: MU, HU, YU, or BAU), level of education (pre-clinical: 1st, 2nd, or 3rd year vs. clinical: 4th, 5th, or 6th year), and the latest Grade Point Average (GPA; ≥3.0 vs. <3.0). Assessment of knowledge regarding HIV transmission involved ten items with an opening question “HIV can be transmitted through”: (1) Sexual relations; (2) Infected syringes and needles; (3) Blood transfusion; (4) Mother-to-child during pregnancy and labor; (5) Mother-to-child via breastfeeding; (6) Handshaking; (7) Mosquito bites; (8) Hugging a person with HIV; (9) Saliva of a person with HIV/AIDS; and (10) Using the same tableware used by a person with HIV/AIDS.

Additional four items were used to assess knowledge on HIV prevention and management as follows: (1) Nowadays, it is possible to prevent HIV transmission from mother to fetus; (2) The chance of HIV infection after exposure could be lowered if given on time after conducting unprotected sexual intercourse; (3) The chance of HIV infection after exposure could be lowered if given on time after a prick from an infected needle; and (4) HIV treatment prolongs the life expectancy of PLWHA. Each of the aforementioned 14 items had the following three options as responses (Yes; No; I do not know).

### 2.4. Assessment of the Stigmatizing Attitude towards PLWHA

The construction of the HIV stigmatization scale (HIV-stigma scale), was based on items adopted from previous relevant studies [41,42,43]. Face validity and content validity of the survey items were checked by the first author and the senior author before survey distribution. The HIV-stigma scale involved 15 items; each was scored using a 5-point Likert scale (highly agree, agree, neutral, disagree, and highly disagree). Internal consistency of the HIV-stigma scale was ensured by Cronbach’s *α* value of 0.863 indicating very good reliability of the scale [44].

For 13 items, the scoring system was as follows: highly disagree (+2), disagree (+1), neutral (zero), agree (−1), and highly agree (−2); and these items included: (1) People with HIV/AIDS got what they deserve; (2) It is hard for me to like people who exposed themselves and society to HIV/AIDS; (3) People with AIDS should be quarantined; (4) Sexual relations should be prohibited for those with HIV/AIDS; (5) Students with HIV/AIDS should be expelled from medical studies; (6) If I would have had HIV/AIDS, I would be ashamed of it; (7) Other students should be notified if one of the medical students is HIV-positive; (8) A physician who is HIV-positive should not be allowed to work even with the appropriate precautions; (9) I believe I have the full right to refuse treating a person with HIV/AIDS; (10) I wish not to treat persons with HIV/AIDS; (11) I would warn other medical staff about a patient’s HIV status even against that patient’s request; (12) if, as an intern, you had to care for a person with HIV/AIDS, would you feel anxious? and (13) I am concerned that working with people who have HIV/AIDS may endanger my health.

For the remaining two items, the score system was reversed as follows: Highly disagree (−2), disagree (−1), neutral (zero), agree (+1), and highly agree (+2); and these items included: (14) I would have a friendship with people with HIV/AIDS; and (15) The professional education I received gave me enough information to confidently work with PLWHA.

The total HIV-stigma score was used to classify the overall stigmatizing attitude towards PLWHA as follows: An HIV-stigma score of (+30 to +16) indicated a highly positive attitude, (+15 to +1) indicated a positive attitude, a score of zero indicated a neutral attitude, a score of (−1 to −15) indicated a negative attitude, and a score of (−16 to −30) indicated an overall highly negative attitude.

### 2.5. Statistical Analysis

All statistical analyses were performed using IBM SPSS Statistics for Windows, Version 22.0. Armonk, NY, US: IBM Corp. Associations between categorical variables were assessed using the chi-squared test, while associations between dichotomous categorical variables and the continuous variable (HIV-stigma scale) were performed using the Mann-Whitney *U* test (M-W). Multinomial regression analyses were used as appropriate and the statistical significance was considered for *p* < 0.050 as the cut-off.

## 3. Results

### 3.1. Characteristics of the Study Sample

The total number of medical students who participated in the study was 1362, with a majority being females (*n* = 780, 57.3%), Jordanians (*n* = 1138, 83.6%), pre-clinical students (*n* = 853, 62.6%), and participants with a latest GPA between 3.5 and 4.0 (*n* = 697, 51.2%, Table 1). The mean age of the respondents was 20 years (median: 21 years, standard deviation (SD): 2.0, interquartile range: 19–22 years, range: 17–31 years). The majority of students were affiliated to the UJ (*n* = 802, 58.9%), followed by JUST (*n* = 219, 16.1%).

Stratified into two categories, the majority of the study respondents were younger than 21 years (*n* = 733, 53.8%), affiliated to medical schools founded before 2000 (*n* = 1021, 75.0%), and had a latest GPA ≥ 3.0 (*n* = 1124, 82.5%, Table 1).

### 3.2. Variable Levels of HIV/AIDS Knowledge Based on the Survey Item

The overall level of HIV/AIDS knowledge among the study participants is shown in (Figure 1). The items with the lowest percentage of correct responses included: (1) HIV can be transmitted through mother to child via breastfeeding (*n* = 556, 40.8%); (2) HIV can be transmitted through saliva of a person with HIV/AIDS (*n* = 580, 42.6%); (3) It is possible to prevent HIV transmission from mother to fetus (*n* = 665, 48.8%); (4) The chance of HIV infection after exposure could be lowered if treatment is given on time after a prick from an infected needle (*n* = 709, 52.1%); (5) HIV can be transmitted through mosquito bites (*n* = 764, 56.1%); (6) The chance of HIV infection after exposure could be lowered if treatment is given on time after conducting unprotected sexual intercourse (*n* = 766, 56.2%); and (7) HIV can be transmitted through using the same tableware used by a person with HIV (*n* = 819, 60.1%). For the remaining items, the percentage of correct responses exceeded 80.0% (Figure 1).

The comparison of HIV/AIDS knowledge per item, based on the level of study is shown in (Table 2). Overall, the clinical students showed a better knowledge regarding HIV transmission compared to the pre-clinical students as illustrated in (Table 2). Females showed a higher percentage of correct responses compared to males for the following item: mother-to-child transmission through breastfeeding (45.6% vs. 34.4%; *p* < 0.001, chi-squared test), while males showed a higher percentage of correct responses to the items: non-transmission through saliva (48.5% vs. 38.2%; *p* < 0.001, chi-squared test), non-transmission through handshaking (95.7% vs. 91.7%; *p* = 0.012, chi-squared test), non-transmission by hugging an infected person (94.0% vs. 90.1%; *p* = 0.014, chi-squared test), and using the same tableware used by an infected person (65.1% vs. 56.4%; *p* < 0.001, chi-squared test).

### 3.3. Analysis of the Stigmatizing Attitude towards PLWHA

Evaluation of the overall stigmatizing attitude among the study respondents showed that about two-thirds of medical students who participated in the study displayed a positive attitude towards PLWHA (positive: *n* = 682; 50.1%; with a mean HIV stigma score of 7.2, and SD of 4.0, and highly positive: *n* = 218; 16.0%; with a mean HIV stigma score of 20.9, and SD of 3.9). A negative attitude was observed among 378 respondents (27.8%); with a mean HIV stigma score of −5.5, and SD of 3.8, and a highly negative attitude was found among 33 respondents only (2.4%); with a mean HIV stigma score of −18.9, and SD of 3.3. Being neutral (having a total HIV-stigma score of zero) was seen among 51 respondents (3.7%, Figure 2).

Multinomial logistic regression analysis was performed to evaluate the possible association between HIV/AIDS knowledge items with the stigmatizing attitude as the dependent variables (positive vs. negative with exclusion of the neutral category), the 14 HIV-knowledge items as the factors and the following as dichotomous covariates (age, sex, nationality, medical school, level of study, and the latest GPA).

This analysis showed that the lack of knowledge (incorrect response compared to correct responses) of the following items was significantly associated with a negative attitude towards PLWHA: (1) HIV can be transmitted through hugging a person with HIV (odds ratio (OR): 9.88, 95% confidence interval (CI): 2.84–34.34, *p* < 0.001); (2) HIV can be transmitted through using the same tableware used by a person with HIV/AIDS (OR: 2.02, 95% CI: 1.37–2.99, *p* < 0.001); (3) HIV can be transmitted through saliva of a person with HIV/AIDS (OR: 1.71, 95% CI: 1.22–2.40, *p* = 0.002); (4) it is possible to prevent HIV transmission from mother to fetus (OR: 2.08, 95% CI: 1.27–3.39, *p* = 0.003), (5) the chance of HIV infection after exposure could be lowered if given on time after a prick from an infected needle (OR: 2.22, 95% CI: 1.32–3.74, *p* = 0.003); and (6) HIV can be transmitted through mosquito bites (OR: 1.52, 95% CI: 1.10–2.10, *p* = 0.012, Table 3).

On the other hand, lack of knowledge for a single item with a borderline *p* value was associated with positive attitude towards PLWHA (HIV can be transmitted from mother to child via breastfeeding (OR: 0.71, 95% CI: 0.51–0.99, *p* = 0.045, Table 3).

### 3.4. Predictors of HIV Stigmatizing Attitude among the Study Respondents Based on HIV-Stigma Scale

To correlate the different respondents’ variables with the stigmatizing attitude towards PLWHA, we conducted multinomial logistic regression analysis. This analysis showed that the pre-clinical students had a significantly more negative attitude towards PLWHA compared to their clinical counterparts (OR: 0.65, 95% CI: 0.43–0.97, *p* = 0.036). For further dissection of this result, direct comparison between the two groups using the HIV-stigma score as a continuous variable showed that the mean HIV-stigma score was +4.1 among the pre-clinical students (SD = 10.1), compared to a mean score of +6.5 among the clinical students (SD = 10.1, *p* < 0.001, M-W, Figure 3A).

Additionally, being affiliated to medical schools that were established before the year 2000 was associated with a significantly more positive attitude towards PLWHA compared to affiliation to the recently founded medical schools in the country (OR: 1.85, 95% CI: 1.42–2.42, *p* < 0.001, Table 3). The mean HIV-stigma score among students affiliated to UJ or JUST as a single group was +5.8 (SD = 10.1) compared to a mean score +2.6 (SD = 9.9, *p* < 0.001, M-W) among MU, HU, YU, or BAU as the second group (Figure 3B).

## 4. Discussion

The concept of HIV stigma is worth a special attention in the MENA region countries for the following reasons: First, previous studies from the region demonstrated the profound lack of knowledge about HIV/AIDS among college students, healthcare workers, as well as among the general public [30,35,36,45,46,47,48].

Second, the public discussion about STIs including HIV infection in the MENA region remains scarce, with socio-cultural sensitivities surrounding such topics. This can contribute to spread of STIs resulting in hidden epidemics [49,50].

Third, the risk groups for HIV acquisition are considered as marginalized, often neglected groups in some MENA countries. In turn this further augments the perceived stigma and discrimination among such groups (MSM, IDUs, FSWs and prisoners) [51,52,53,54,55].

Finally, previous studies among college students in the MENA region have shown severe gaps in knowledge regarding STIs in general and HIV/AIDS in particular [33,56,57]. Therefore, the assessment of HIV/AIDS knowledge and the stigmatizing attitude towards PLWHA appears invaluable as an initial step to tailor well-informed actions. In turn, this approach can be helpful to tackle the increase in HIV infections in the MENA region [58].

To the best of our knowledge, the current study is the first to evaluate HIV/AIDS knowledge and to emphasize the possible stigmatizing attitude towards PLWHA among medical students in Jordan. The results of this study showed variable levels of HIV/AIDS knowledge depending on the survey item and revealed the presence of significant defects. These defects involved HIV/AIDS knowledge aspects such as HIV transmission and prevention. Only 41% of the study participants were aware that HIV can be transmitted from mother-to-child through breastfeeding, and only 49% were aware that it is possible to prevent mother-to-child transmission of HIV. Such a result can be considered as a major gap in HIV/AIDS knowledge with subsequent negative impact on the feasible preventive measures [55].

A previous study from Hanoi, Vietnam, showed that the quality of prevention of vertical HIV transmission can be enhanced by improving the skills of healthcare workers, including lessening the negative attitude due to fear of infection. Such improvements can enable healthcare workers to provide a greater support for PLWHA [59]. In line with the results of this study, unsatisfactory levels of HIV/AIDS knowledge were reported in the MENA region among college students in the United Arab Emirates and Yemen, medical, and dental students in Saudi Arabia, and healthcare workers in Egypt [35,56,57,60,61].

A previous study that was conducted among nurses in Jordan showed a lower level of HIV/AIDS knowledge in association with a negative attitude towards PLWHA resulting from fear of infection and social stigma [62]. Despite the low prevalence of HIV and STIs in the country, PLWHA in Jordan face several challenges in relation to quality of care besides the problems of unemployment, low income, and non-disclosure status; all of which were related to suboptimal health-related quality of life [63,64].

Another finding in this study that could hinder the prevention of HIV transmission in healthcare settings was related to lack of knowledge regarding postexposure prophylaxis (PEP). Only 52% of the respondents were aware that the risk of HIV acquisition can be lowered following needle-stick injury if PEP is initiated. Occupational exposure to HIV among other bloodborne pathogens poses a continuous risk to healthcare workers; hence, medical students’ knowledge regarding the value of PEP, and its practical implementation should be an area of special attention in medical education. This is mainly relevant in light of the frequency of needle-stick injuries in clinical practice [65,66].

In the current study, the major finding was that a majority of the study participants held a positive attitude towards PLWHA. However, the results also pointed to the common prevalence of the negative attitude that was found in approximately one-third of medical students who participated in this study. Such a negative attitude can represent a major constraint in the efforts needed to control HIV/AIDS infections in the region [61]. Previous studies from the MENA region showed that a majority of participants held a negative stigmatizing attitude towards PLWHA [32,67]. Dental students in Saudi Arabia showed negative attitudes involving fear of accidental exposure, and reluctance to treat PLWHA [68]. A recent study among female college students in three Arab countries (Bahrain, Jordan, and Kuwait) attributed such a negative attitude to taboos surrounding sexual topics in the region [69].

The finding of a correlation between lack of HIV/AIDS knowledge with a negative stigmatizing attitude towards PLWHA appears fathomable since accurate information about virus transmission can result in less fear and avoidance attitude [70]. The current finding can point to the lack of HIV/AIDS knowledge as a main driver of the stigmatizing attitude among the study participants. This result affirms the continuous need to focus on delivery of proper educational messages to reduce prejudice and stigma, which can be helpful at a society level as well, considering that medical students can form a core group with a leading role in community education and service [71].

An additional result of this study was the finding that the pre-clinical students showed more negative attitude towards PLWHA compared to the clinical students. This might be related to the gradual acquisition of knowledge and skills over their years of study, which was evident by the results of this study for the majority of HIV knowledge items. Higher levels of knowledge among clinical students compared to their pre-clinical counterparts were also reported in recent studies addressing HPV knowledge among medical and dental students in Jordan and Saudi Arabia [33,34,72]. A similar pattern was observed among medical students in Israel and Malaysia and was attributed to the progressive exposure to more medical information (including those pertinent to HIV/AIDS) by advancing through curriculum years [41,73]. In spite of this improvement in HIV/AIDS knowledge observed with advancement in curriculum years, the overall level of knowledge was not satisfactory even among clinical students and this should be addressed through revision of the current medical curricula in the country.

An interesting finding in this study was related to the independent correlation between the affiliation to the recently established medical schools in the country with a more negative attitude towards PLWHA. This result might shed light on the need for incorporation of improved public health training in the medical curricula among such recently established medical schools in the country. However, this result should be interpreted with caution bearing in mind the lower number of participants from the recently established medical schools in the country.

Study limitations included the inherent bias of the snowball sampling approach, the predominance of participants from two medical schools (UJ and JUST, which might be related to the higher number of students enrolled in these two universities that are considered the largest medical schools in the country), and the possibility that the respondents could have answered in a way they believe to be suitable for the researchers.

## 5. Conclusions

The majority of medical students who participated in this study displayed a positive attitude towards PLWHA; however, several defects in HIV/AIDS knowledge were detected. In addition, negative attitude towards PLWHA was observed in about one-third of the participants. Thus, refined educational curricula and clinical training programs are recommended in the country to raise the levels of HIV/AIDS knowledge and to improve the capabilities and preparedness of medical students to manage PLWHA. The improvements in skills and acquisition of adequate HIV/AIDS knowledge can be reflected on the society considering the important role of students in public service and in raising the societal awareness of detrimental effects of HIV stigma and discrimination.

## Figures and Tables

**Figure 1 ijerph-19-00745-f001:**
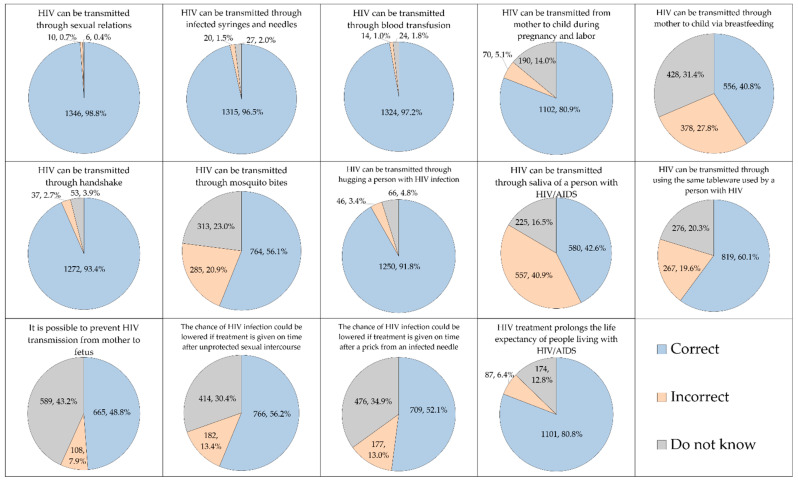
The participants’ knowledge on HIV transmission, management and prevention. HIV: Human immunodeficiency virus; AIDS: Acquired immunodeficiency syndrome.

**Figure 2 ijerph-19-00745-f002:**
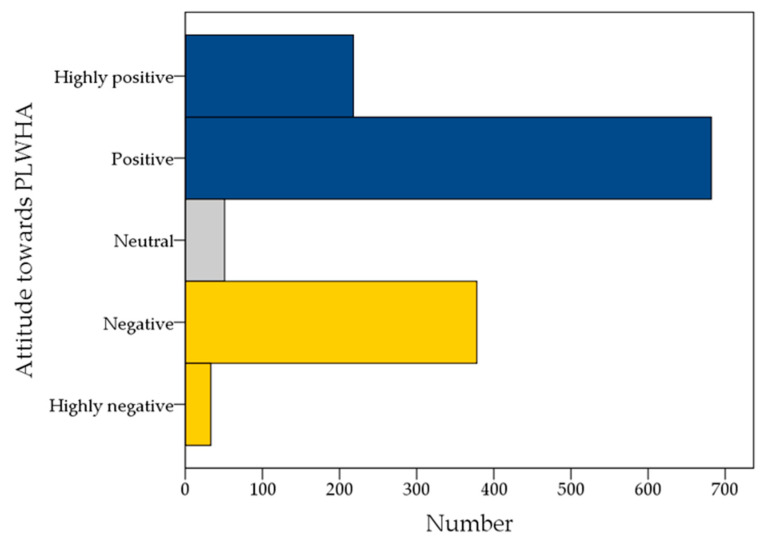
The overall attitude of the study respondents towards people living with HIV/AIDS (PLWHA) based on HIV stigma scale. An HIV-stigma score of +30 to +16 indicated highly positive attitude, +15 to +1 indicated positive attitude, a score of zero indicated a neutral attitude, −1 to −15 indicated a negative attitude, and −16 to −30 indicated an overall highly negative attitude. HIV: Human immunodeficiency virus; AIDS: Acquired immunodeficiency syndrome.

**Figure 3 ijerph-19-00745-f003:**
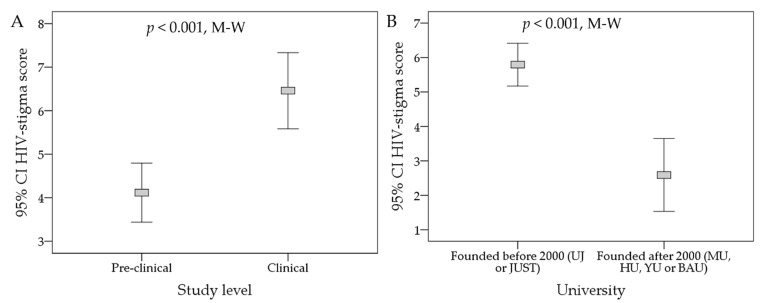
Factors that were correlated with significant differences in HIV-stigma score among the study participants. (**A**) Comparison was based on the study level of the participants (pre-clinical vs clinical students); (**B**) Comparison was based on the time the medical school was founded (before the year 2000 vs. following the year 2000). HIV: Human immunodeficiency virus, UJ: the University of Jordan; JUST: Jordan university of science and technology; MU: Mutah University; HU: the Hashemite University; YU: Yarmouk University; BAU: Al-Balqa Applied University; M-W: Mann-Whitney *U* test, 95% CI: 95% confidence interval of the mean; the grey rectangles in error bars represent the mean.

**Table 1 ijerph-19-00745-t001:** Characteristics of the study participants.

Variable	Category	Number	Percentage
Sex	Male	582	42.7%
	Female	780	57.3%
Nationality	Jordanian	1138	83.6%
	Non-Jordanian ^2^	222	16.3%
	Preferred not to say	2	0.1%
University	University of Jordan	802	58.9%
	JUST ^3^	219	16.1%
	Mutah University	75	5.5%
	Hashemite University	93	6.8%
	Al-Balqa’ Applied University	74	5.4%
	Yarmouk University	99	7.3%
Curriculum year	1st year	266	19.5%
	2nd year	343	25.2%
	3rd year	244	17.9%
	4th year	218	16.0%
	5th year	164	12.0%
	6th year	127	9.3%
Level of study	Pre-clinical	853	62.6%
	Clinical	509	37.4%
Latest GPA ^1^	3.50–4.00	697	51.2%
	3.00–3.49	427	31.4%
	2.50–2.99	199	14.6%
	2.00–2.49	36	2.6%
	Less than 2.00	3	0.2%

^1^ GPA: Grade point average; ^2^ Non-Jordanian: Belonged to 15 different nationalities with a majority from Iraq (*n* = 71), Kuwait (*n* = 60), Palestine (*n* = 39), and Syria (*n* = 21); ^3^ JUST: Jordan University for Science and Technology.

**Table 2 ijerph-19-00745-t002:** Comparison of HIV knowledge among pre-clinical vs. clinical students.

HIV Knowledge Item	Level	Pre-Clinical	Clinical	*p* Value ^3^
	Response ^1^	N ^2^ (%)	N (%)	
HIV can be transmitted through sexual relations	Correct	838 (98.2)	508 (99.8)	0.03239
	Incorrect	9 (1.1)	1 (0.2)	
	I don’t know	6 (0.7)	0	
HIV can be transmitted through infected syringes and needles	Correct	809 (94.8)	506 (99.4)	0.00005
	Incorrect	19 (2.2)	1 (0.2)	
	I don’t know	25 (2.9)	2 (0.4)	
HIV can be transmitted through blood transfusion	Correct	820 (96.1)	504 (99.0)	0.00329
	Incorrect	14 (1.6)	0	
	I don’t know	19 (2.2)	5 (1.0)	
HIV can be transmitted from mother to child during pregnancy and labor	Correct	626 (73.4)	476 (93.5)	<0.00001
	Incorrect	56 (6.6)	14 (2.8)	
	I don’t know	171 (20.0)	19 (3.7)	
HIV can be transmitted through mother to child via breastfeeding	Correct	360 (42.2)	196 (38.5)	0.00005
	Incorrect	202 (23.7)	176 (34.6)	
	I don’t know	291 (34.1)	137 (26.9)	
HIV can be transmitted through handshake	Correct	783 (91.8)	489 (96.1)	0.00155
	Incorrect	33 (3.9)	4 (0.8)	
	I don’t know	37 (4.3)	16 (3.1)	
HIV can be transmitted through mosquito bites	Correct	429 (50.3)	335 (65.8)	<0.00001
	Incorrect	200 (23.4)	85 (16.7)	
	I don’t know	224 (26.3)	89 (17.5)	
HIV can be transmitted through hugging a person with HIV infection	Correct	759 (89.0)	491 (96.5)	<0.00001
	Incorrect	43 (5.0)	3 (0.6)	
	I don’t know	51 (6.0)	15 (2.9)	
HIV can be transmitted through saliva of a person with HIV/AIDS	Correct	348 (40.8)	232 (45.6)	0.21665
	Incorrect	358 (42.0)	199 (39.1)	
	I don’t know	147 (17.2)	78 (15.3)	
HIV can be transmitted through using the same tableware used by a person with HIV	Correct	480 (56.3)	339 (66.6)	0.00001
	Incorrect	200 (23.4)	67 (13.2)	
	I don’t know	173 (20.3)	103 (20.2)	
It is possible to prevent HIV transmission from mother to fetus	Correct	333 (39.0)	332 (65.2)	<0.00001
	Incorrect	79 (9.3)	29 (5.7)	
	I don’t know	441 (51.7)	148 (29.1)	
The chance of HIV infection after exposure could be lowered if treatment is given on time after conducting unprotected sexual intercourse	Correct	432 (50.6)	334 (65.6)	<0.00001
	Incorrect	138 (16.2)	44 (8.6)	
	I don’t know	283 (33.2)	131 (25.7)	
The chance of HIV infection after exposure could be lowered if treatment is given on time after a prick from an infected needle	Correct	386 (45.3)	323 (63.5)	<0.00001
	Incorrect	135 (15.8)	42 (8.3)	
	I don’t know	332 (38.9)	144 (28.3)	
HIV treatment prolongs the life expectancy of people living with HIV/AIDS	Correct	659 (77.3)	442 (86.8)	0.00003
	Incorrect	70 (8.2)	17 (3.3)	
	I don’t know	124 (14.5)	50 (9.8)	

^1^ Response: The percentage of correct responses represents the number of respondents who answered yes to the correct statements or no to the incorrect statements; ^2^ N: Number; ^3^ *p* Value: Calculated using the chi-squared test.

**Table 3 ijerph-19-00745-t003:** Results of multinomial regression analysis of factors associated with a positive attitude towards PLWHA among the study participants.

HIV Knowledge Item (Negative HIV Stigma Compared to Positive Attitude as the Reference)	OR (95% CI) ^2^	*p* Value
HIV can be transmitted through sexual relations (incorrect response)	6.84 (0.71–65.77)	0.096
I don’t know	2.31 (0.24–22.41)	0.470
Correct response (reference)	.	.
HIV can be transmitted through infected syringes and needles (incorrect response)	2.61 (0.71–9.58)	0.150
I don’t know	0.67 (0.22–2.05)	0.478
Correct response (reference)	.	.
HIV can be transmitted through blood transfusion (incorrect response)	1.72 (0.26–11.40)	0.575
I don’t know	0.54 (0.16–1.85)	0.330
Correct response (reference)	.	.
HIV can be transmitted from mother to child during pregnancy and labor (incorrect response)	1.51 (0.84–2.71)	0.173
I don’t know	0.99 (0.65–1.49)	0.944
Correct response (reference)	.	.
HIV can be transmitted from mother to child via breastfeeding (incorrect response)	0.71 (0.51–0.99)	0.045 *
I don’t know	0.85 (0.62–1.17)	0.327
Correct response (reference)	.	.
HIV can be transmitted through handshaking (incorrect response)	0.34 (0.10–1.13)	0.077
I don’t know	0.82 (0.33–2.05)	0.676
Correct response (reference)	.	.
HIV can be transmitted through mosquito bites (incorrect response)	1.52 (1.10–2.10)	0.012 *
I don’t know	1.04 (0.74–1.47)	0.815
Correct response (reference)	.	.
HIV can be transmitted through hugging a person with HIV (incorrect response)	9.88 (2.84–34.34)	<0.001 *
I don’t know	1.67 (0.76–3.64)	0.202
Correct response (reference)	.	.
HIV can be transmitted through saliva of a person with HIV/AIDS (incorrect response)	1.71 (1.22–2.40)	0.002 *
I don’t know	1.02 (0.66–1.57)	0.936
Correct response (reference)	.	.
HIV can be transmitted through using the same tableware used by a person with HIV/AIDS (incorrect response)	2.02 (1.37–2.99)	<0.001 *
I don’t know	1.56 (1.09–2.24)	0.015 *
Correct response (reference)	.	.
Nowadays, it is possible to prevent HIV transmission from mother to fetus (incorrect response)	2.08 (1.27–3.39)	0.003 *
I don’t know	1.05 (0.78–1.40)	0.769
Correct response (reference)	.	.
The chance of HIV infection after exposure could be lowered if given on time after conducting unprotected sexual intercourse (incorrect response)	1.19 (0.73–1.94)	0.479
I don’t know	1.38 (0.95–1.98)	0.088
Correct response (reference)	.	.
The chance of HIV infection after exposure could be lowered if given on time after a prick from an infected needle (incorrect response)	0.90 (0.55–1.47)	0.663
I don’t know	0.83 (0.58–1.19)	0.310
Correct response (reference)	.	.
The chance of HIV infection after exposure could be lowered if given on time after a prick from an infected needle (incorrect response)	2.22 (1.32–3.74)	0.003 *
I don’t know	1.43 (0.97–2.10)	0.075
Correct response (reference)	.	.
**Covariates**		
Age	1.23 (0.80–1.90)	0.350
Sex	0.81 (0.62–1.05)	0.113
Nationality	1.05 (0.73–1.51)	0.801
Medical school	1.54 (1.14–2.08)	0.005
Level of study	0.82 (0.52–1.30)	0.406
Latest GPA ^1^	1.19 (0.83–1.71)	0.341

^1^ GPA: Grade point average; ^2^ OR: Odds ratio, CI: Confidence interval. HIV: Human immunodeficiency virus. Significant differences are marked with an asterisk.

## Data Availability

The data that support the findings of this study are available from the corresponding author (M.S.) upon a reasonable request.

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
