# Peer review of "HIV Knowledge and Stigmatizing Attitude towards People Living with HIV/AIDS among Medical Students in Jordan"

_ijerph, 2022, doi:10.3390/ijerph19020745_

Round 1

Reviewer 1 Report

The study examined HIV/AIDS knowledge and stigmatizing attitude among medical students in Jordan. Results are informative and important in identifying HIV/AIDS knowledge gaps and stigma among medical students and in informing the need of public health education. The manuscript is in general well-written. Below are few minor comments to further improve the manuscript:

  • As stigmatizing attitude wards PLWHA is a continuous variable, the authors may consider presenting the mean score in addition to the frequencies of each response (section 3.3)
  • The authors analysed how clinical and pre-clinical students (and other categorical variables such as affiliation) differ in the mean stigmatizing attitude. However, instead of using multinomial logistic regression, I wonder why the t-tests were not used to directly compare the mean levels between different groups (section 3.4)?

Author Response

The study examined HIV/AIDS knowledge and stigmatizing attitude among medical students in Jordan. Results are informative and important in identifying HIV/AIDS knowledge gaps and stigma among medical students and in informing the need of public health education. The manuscript is in general well-written. Below are few minor comments to further improve the manuscript:

Response: We are deeply grateful for the positive appraisal of the manuscript.

As stigmatizing attitude wards PLWHA is a continuous variable, the authors may consider presenting the mean score in addition to the frequencies of each response (section 3.3)

Response: We would like to thank the reviewer for this important suggestion and accordingly, we added the mean scores and standard deviations besides the frequencies (Page 6 of the revised highlighted manuscript).

The authors analysed how clinical and pre-clinical students (and other categorical variables such as affiliation) differ in the mean stigmatizing attitude. However, instead of using multinomial logistic regression, I wonder why the t-tests were not used to directly compare the mean levels between different groups (section 3.4)?

Response: We are thankful for this important comment. We used multinomial logistic regression to control for possible confounders (age, sex, nationality, medical school, level of study, and the latest GPA).

Reviewer 2 Report

Overall, a well-done and necessary study. 

Suggestions to tighten the introduction and discussion - in their current forms, they are overlong and this impacts negatively on the readability of the manuscript as a whole. Efforts to focus the introduction on the state of HIV stigma and services in the MENA region and Jordan specifically, as well as focusing the discussion on factors associated with HIV knowledge and stigma, and possible measures to improve these, would strengthen the paper. 

Author Response

Overall, a well-done and necessary study. 

Suggestions to tighten the introduction and discussion - in their current forms, they are overlong and this impacts negatively on the readability of the manuscript as a whole. Efforts to focus the introduction on the state of HIV stigma and services in the MENA region and Jordan specifically, as well as focusing the discussion on factors associated with HIV knowledge and stigma, and possible measures to improve these, would strengthen the paper.  

Response: We are deeply grateful for the positive appraisal of the manuscript, and regarding the length of the introduction and discussion, we feel that the current sections can provide a clear context regarding the rationale behind conducting the study, which would help the readers, particularly from outside the Middle East and North Africa, to grasp the importance of addressing the issue of HIV stigma in the region and to appreciate the need for special attention regarding this topic. Thus, we prefer to keep the details provided in the introduction and discussion in the current form.